# Clinical Implication of Bilateral and Unilateral Multifocality in Papillary Thyroid Carcinoma: A Propensity Score-Matched Study

**DOI:** 10.3390/cancers15143596

**Published:** 2023-07-13

**Authors:** Youngmin Kim, Solji An, Joonseon Park, Ja Seong Bae, Jeong Soo Kim, Kwangsoon Kim

**Affiliations:** Department of Surgery, College of Medicine, The Catholic University of Korea, Seoul 06591, Republic of Korea; ym.e.kim@gmail.com (Y.K.); solji1130@hanmail.net (S.A.); joonsunny@naver.com (J.P.); jaseong@gmail.com (J.S.B.); btskim@catholic.ac.kr (J.S.K.)

**Keywords:** bilaterality, unilateral multifocality, papillary thyroid carcinoma, disease-free survival, propensity score matching

## Abstract

**Simple Summary:**

Multifocality is often found in PTC. There are contradicting evidences regarding clinical implications of cancer laterality in multifocal PTC. The current study compared the clinicopathological characteristics and the long-term oncological outcomes of bilateral and unilateral multifocal PTC. Patients with bilateral PTC were more likely to present with aggressive features, including high tumor stage, cervical LN metastasis, gross ETE, and lymphatic and perineural invasion. On the multivariate analysis using Cox regression, gross ETE, perineural invasion, LN metastasis, N1b stage, and RAI therapy were significant risk factors of recurrence. However, bilaterality was not associated with an increased risk of recurrence. In order to reduce selection bias, a subgroup analysis of patients who underwent total thyroidectomy was conducted using propensity score matching. After matching, unilateral multifocality, vascular invasion, and RAI therapy were significant risk factors of recurrence. Patients with unilateral multifocal PTC showed significantly poorer DFS than those with bilateral PTC on the Kaplan-Meier analysis.

**Abstract:**

Papillary thyroid cancer (PTC) is commonly characterized by multifocality, which is associated with aggressive features and a less favorable prognosis. The current study aimed to compare the clinicopathologic characteristics and long-term oncological outcomes of bilateral and unilateral multifocal PTC. The medical records of 1745 patients with multifocal PTC who underwent thyroid surgery at Seoul St. Mary’s Hospital were retrospectively reviewed. The clinicopathological characteristics and recurrence rates were compared based on cancer laterality. Further, 357 patients who underwent total thyroidectomy were matched to investigate the recurrence risk and disease-free survival (DFS). Before propensity score matching (PSM), there was no significant difference in the recurrence rate between the bilateral and unilateral multifocal PTC groups. Cancer laterality was not a predictor of DFS based on the Cox regression analyses. However, after PSM, unilateral multifocality was associated with a significantly high risk of recurrence. Similarly, unilateral multifocality was associated with a significantly poor DFS based on the Kaplan–Meier analysis. Compared with bilateral PTC, unilateral multifocal PTC was associated with a poor DFS. A comprehensive preoperative examination should be performed to detect multifocality before the initial surgical intervention for optimal treatment. Postoperative short-term follow-up is recommended for unilateral multifocal PTC for recurrence surveillance.

## 1. Introduction

Over the years, the incidence of thyroid cancer has been continuously increasing particularly in developed countries, partly due to advancements in imaging technology and increasing accuracy of pathologic evaluation [1,2]. Papillary thyroid cancer (PTC) is the most prevalent subtype of thyroid carcinoma, accounting for up to 89.1% of all diagnosed cases [3].

The incidence of multifocal PTC ranges from 18% to 87% [4,5,6]. It is not uncommon for PTC to present as a multifocal disease at diagnosis. Moreover, multifocal PTC is often incidentally diagnosed during postoperative pathologic evaluation. Multifocality is defined as the presence of two or more distinct malignant foci in the thyroid gland. According to the location of the malignant foci, multifocality can be divided into either unilateral—if multiple foci are limited to a single lobe—or bilateral—if malignant foci are found in two lobes. However, there is still a debate whether to consider multiple foci as a characteristic of independent synchronous malignancies or intraglandular metastases from a single focus [4,7].

Several studies have shown that bilaterality is an independent predictor of lymph node metastasis in PTC. Further, it is related to a more aggressive disease, with an increased risk of gross extrathyroidal extension (ETE), positive resection margin, locoregional recurrence, and distant metastasis [7,8,9]. According to the National Comprehensive Cancer Network (NCCN) guidelines, total thyroidectomy (TT) is the recommended surgical extent for bilateral PTC. If macroscopic multifocality is diagnosed postoperatively upon pathologic examination, a complete thyroidectomy should be performed [10].

On the contrary, some studies have revealed that although bilateral PTC is associated with more aggressive features, patients with this condition do not necessarily have a poor clinical outcome [11,12]. Further, compared with TT, unilateral lobectomy may be sufficient for treating low-risk unilateral multifocal PTC without cervical LN metastasis, with no negative impact on recurrence-free survival [13]. Thus, the negative effect of multifocality on clinical course remains unclear.

The current study aimed to compare the clinicopathological characteristics and long-term oncological outcomes of unilateral multifocal and bilateral PTC via a propensity score-matched analysis to reduce selection bias.

## 2. Materials and Methods

### 2.1. Patients

Between March 2008 and June 2014, 4591 patients diagnosed with PTC underwent a thyroidectomy at Seoul St. Mary’s Hospital. In total, 139 patients were excluded due to insufficient data (*n* = 84) and loss to follow-up (*n* = 55). The medical records of the remaining 4452 patients were reviewed retrospectively. Then, 1745 patients with multifocal PTC were included in the analysis. Moreover, 1002 and 743 patients had bilateral and unilateral multifocal PTC. All patients with bilateral disease underwent total thyroidectomy (TT), and 371 (49.9%) of 743 patients with unilateral multifocal disease underwent TT. TNM classification was based on the eighth edition of the AJCC TNM staging system [14]. Testing for BRAF^V600E^ mutation was carried out via real-time PCR of tumor specimen acquired by either fine-needle biopsy or surgical resection, as well as BRAF^V600E^-specific staining of tumors. A positive result was considered as positive for BRAF mutation. The average follow-up duration was 105.4 ± 22.2 (range: 55–139) months.

This study was conducted in accordance with the Declaration of Helsinki (revised in 2013). It was approved by the Institutional Review Board of Seoul St. Mary’s Hospital, Catholic University of Korea (no: KC22RISI0682). The need for informed consent was waived due to the retrospective nature of this study.

### 2.2. Follow-Up Assessment

Postoperative care and follow-up evaluation were conducted according to the American Thyroid Association (ATA) management guidelines for differentiated thyroid cancer [15]. During regular follow-up evaluations, all patients underwent a physical examination, blood tests, including serum thyroid function, thyroglobulin and anti-thyroglobulin antibody assessment, and neck ultrasonography at 3, 6, and 12 months after surgery and annually after the first year. Radioactive iodine (RAI) ablation was performed 6–8 weeks after the surgery in patients who underwent TT and those who are at intermediate or high risk according to the risk stratification system of the ATA management guidelines. A whole-body scan was performed 5–7 days after RAI ablation. To determine the location and extent of recurrence, patients suspected with recurrence underwent additional diagnostic imaging tests, including a computed tomography scan, positron emission tomography/computed tomography scan, and/or whole-body RAI scan. If a structural recurrence appeared on imaging studies, either ultrasound-guided fine-needle aspiration/core needle biopsy or surgical excision was performed for pathologic confirmation. 

### 2.3. Primary and Secondary Endpoint

The primary endpoint was disease-free survival (DFS) between the unilateral multifocal and bilateral PTC groups after propensity score matching (PSM). The secondary endpoint was clinicopathological characteristics between the two groups before and after PSM.

### 2.4. Statistical Analysis

Continuous variables were analyzed with the Student’s *t*-test and were presented as means and standard deviations. Categorical variables were assessed with the Fisher’s exact test and were expressed as numbers and percentages. Univariate and multivariate Cox regression analyses were performed to validate the predictors of DFS, and hazard ratios (HRs) with 95% confidence intervals (CIs) were calculated. The Kaplan–Meier method with the log-rank test was used to compare the DFS.

To control potential confounding factors, the individual patient propensity scores were calculated via logistic regression analysis. Next, patients with unilateral multifocal PTC were matched to those with bilateral PTC at a 1:1 ratio using the propensity scores. Then, the DFS and long-term oncological outcomes were compared between the matched unilateral multifocal and bilateral PTC groups. After PSM, the DFS predictors were validated using univariate and multivariate Cox regression analyses, similar to the approach before matching. A *p* value of <0.05 was considered statistically significant. All statistical analyses were performed using the Statistical Package for the Social Sciences software (version 24.0; IBM Corp., Armonk, NY, USA).

## 3. Results

### 3.1. Comparison of Bilateral and Unilateral Multifocal PTC

Table 1 shows the baseline clinicopathological characteristics of bilateral and unilateral multifocal PTC. The mean age of the bilateral PTC group was significantly higher than that of the unilateral multifocal PTC group, while the two groups did not show a difference in sex. All patients with bilateral PTC underwent TT with central lymph node dissection and/or modified radical neck dissection. Meanwhile, 50.1% of patients with unilateral multifocal disease underwent thyroid lobectomy with central lymph node dissection only. On pathological examination, patients with bilateral PTC had significantly larger tumors than those with unilateral multifocal PTC (1.2 ± 0.9 cm vs. 0.9 ± 0.6 cm, *p* < 0.001). Moreover, the bilateral PTC group more commonly presented with gross ETE (9.5% vs. 3.9%, *p* < 0.001), lymphatic invasion (36.6% vs. 26.1%, *p* < 0.001), and perineural invasion (3.9% vs. 1.3%, *p* = 0.001) than the unilateral multifocal PTC group. Patients with bilateral PTC had a higher number of harvested lymph nodes and metastatic lymph nodes than those with unilateral multifocal PTC (19.1 ± 21.4 vs. 13.2 ± 16.0, *p* < 0.001 and 3.9 ± 6.3 vs. 2.4 ± 4.5, *p* < 0.001, respectively). Accordingly, the bilateral PTC group had a higher proportion of patients with more advanced T and N stages than the unilateral multifocal PTC group (*p* < 0.001, both). Although not statistically significant, the bilateral PTC group had a higher proportion of patients with M1 disease than the unilateral multifocal group (0.4% vs. 0.1%, *p* = 0.402). Approximately 72.5% of patients in the bilateral PTC group and 36.2% in the unilateral multifocal group received RAI therapy after surgical treatment (*p* < 0.001). However, the two groups did not significantly differ in terms of recurrence (4.3% vs. 4.4%, *p* = 0.906).

### 3.2. Univariate and Multivariate Analyses of the Risk Factors of Recurrence before PSM

In the univariate analysis, male sex, tumor size > 1 cm, gross ETE, lymphatic, vascular invasion, and perineural invasion, LN metastasis, T stage of ≥2, N1 stage, M1 disease, and RAI therapy were considered the risk factors of recurrence (Table 2). Based on the multivariate analysis, only gross ETE, perineural invasion, LN metastasis, N1b stage, and RAI therapy were significantly associated with recurrence. However, bilaterality was not a significant risk factor of recurrence.

### 3.3. Comparison of Patients with Bilateral and Unilateral Multifocal PTC Who Underwent TT

Table 3 shows the baseline clinicopathological characteristics of bilateral and unilateral multifocal PTC in patients who underwent TT before and after PSM. All patients with bilateral PTC underwent TT. Meanwhile, 371 of 743 patients with unilateral multifocal PTC underwent TT. Before matching, the bilateral PTC group had a significantly higher proportion of patients with gross ETE than the unilateral multifocal PTC group (9.5% vs. 5.7%, *p* = 0.028). Similarly, the bilateral PTC group had a higher proportion of patients with T2 and T3 disease than the unilateral multifocal group (6.8% vs. 4.3% and 10.6% vs. 5.7%, respectively, *p* = 0.015). After PSM, the unilateral multifocal PTC group had a higher recurrence rate than the bilateral PTC group (2.2% vs. 5.9%, *p* = 0.021).

### 3.4. Univariate and Multivariate Analyses of Recurrence Risk Factors after PSM

Table 4 depicts the risk factors of recurrence after PSM based on the univariate and multivariate analyses. Based on the univariate analysis, tumor size > 1 cm, gross ETE, unilateral multifocality, lymphatic/vascular/perineural invasion, lymph node positivity, T3b and T4b stages, N1a and N1b stages, and RAI therapy were significant risk factors of recurrence. Based on the multivariate analysis, only unilateral multifocality (hazard ratio (HR): 2.664, 95% CI: 1.180–6.017; *p* = 0.018), vascular invasion (HR: 3.839, 95% CI: 1.331–11.073; *p* = 0.013), and RAI therapy (HR: 12.124, 95% CI: 1.640–89.630; *p* = 0.014) were significant predictors of recurrence. In the Kaplan–Meier analysis, patients with unilateral multifocal disease had a significantly lower DFS than those with bilateral disease (*p* = 0.014, Figure 1).

## 4. Discussion

Previous studies have reported that bilateral PTC is a predictor of aggressive disease and an independent risk factor of recurrence [8,9,16]. However, in the current study, unilateral multifocality, rather than bilaterality, was associated with a high risk of recurrence.

PTC is the most prevalent subtype of thyroid malignancies [3]. Generally, PTC is a relatively indolent tumor with an excellent long-term prognosis. Its overall survival rate is 90–95% [17]. Multifocal PTC is common, with an incidence rate of 18–87%, based on the diagnostic approach [4,5,6]. According to the ATA guidelines, multifocal PTC is associated with a low risk for structural disease recurrence unless it is accompanied by other high-risk features such as ETE, LN metastasis, and vascular invasion [15]. Nonetheless, several studies have reported that multifocality is associated with a high risk of contralateral disease and that bilateral multifocality is a predictor of LN metastasis and poorer clinical outcome, including locoregional recurrence and distant metastasis [18,19,20,21,22,23]. Feng et al. showed that both multifocality and bilaterality were associated with aggressive features. However, multifocality was the only predictor of recurrence [24]. Similarly, Yan et al. found that bilateral multifocality is associated with aggressive features such as large tumor size, ETE, and lymph node metastasis, but not with prognosis [25]. Therefore, whether multifocality or bilaterality can be a prognostic factor remains controversial.

Results showed that patients with bilateral PTC had significantly larger tumors and more extensive lymph node involvement and advanced T and N stages compared with those with unilateral multifocal PTC. Moreover, bilateral disease was more likely characterized by aggressive features, including gross ETE and lymphatic and perineural invasion. These findings are in accordance with those of previous reports showing that bilateral PTC is associated with more aggressive clinicopathological features compared with unilateral multifocal disease [11,24,25]. Interestingly, the incidence of BRAF^V600E^ mutation and the recurrence rate did not significantly differ between the bilateral and the unilateral multifocal PTC groups, which was in contrast to previously reported outcomes [26,27].

Patients with bilateral PTC are routinely treated with TT. Meanwhile, patients with unilateral multifocal PTC undergo either TT or lobectomy. The surgical extent may have an effect on locoregional disease control and may subsequently affect long-term outcomes. Hence, patients who underwent TT were analyzed. Approximately 49.9% of patients with unilateral multifocal disease who underwent TT were included in the research. Results showed that the bilateral PTC group had a significantly larger tumor size and higher frequency of gross ETE, thereby leading to a greater proportion of patients with a more advanced T stage. Otherwise, the two groups did not significantly differ in terms of other aggressive features, such as aggressive PTC subtypes, cervical LN metastasis, BRAF^V600E^ positivity, and lymphatic, vascular, and perineural invasion.

We conducted PSM to further reduce selection bias. After matching, unilateral multifocality, vascular invasion, and RAI therapy were the only significant risk factors of recurrence. Interestingly, unilateral multifocality was a predictor of recurrence based on the multivariate analysis, with an HR of 2.664. This result is in contrast to that of previous studies showing that bilateral PTC is a risk factor of recurrence [8,9,16].

PSM yielded rather an unexpected outcome. Before matching, the bilateral and the unilateral multifocal PTC groups did not show a significant difference in recurrence, which became significant once PSM was conducted. It suggests that the clinicopathologic characteristics, such as tumor size and gross extrathyroidal extension, play an important role in determining the oncological outcome. Therefore, it is important to correct for the possibly confounding factors in order to make an accurate comparison of cancer laterality regarding prognosis. Our findings showed that the surgical extent for unilateral multifocal PTC should be planned cautiously. Postoperatively, patients with unilateral multifocal PTC must be closely monitored with short-term follow-up for the early detection of recurrent disease.

A molecular analysis of multifocal PTCs by Bansal and colleagues revealed that PTCs arising from different molecular origins, representing multiple synchronous primary tumors (MSPTs), frequently appear in different lobes, while the majority of tumors that share the same mutation status were located in the same lobe [28]. Similarly, Cai et al. have shown that unilateral multifocality and bilateral multifocality should be considered as two molecularly separate entities. They found that unilateral multifocal disease is associated with an increased risk of central neck metastasis compared with bilateral disease (single lesion in each lobe) [29]. Together, these findings suggest that unilateral multifocal PTC may be more aggressive in nature because it arises through intraglandular tumor dissemination rather than synchronous development of independent primary tumors.

Further, vascular invasion was a strong predictor of recurrence in our study, with an HR of 3.839. Reilly et al. showed that vascular invasion is associated with aggressive tumor factors, such as lymphatic and capsular invasion, ETE, and lymph node metastasis, in papillary cancer [30]. Our findings are in accordance with those of previous studies showing that vascular invasion is associated with a high risk of recurrence and poor prognosis in PTC [30,31].

RAI therapy is recommended for intermediate- and high-risk patients who underwent TT or near TT, according to the current ATA guidelines [15]. However, the effect of RAI therapy in improving prognosis remains controversial, with a RAI-refractory PTC rate of 20–30% [32]. Tang J. et al. performed a retrospective study using the SEER program. Results showed that RAI therapy improved disease-specific survival in patients with a tumor measuring <2 cm and distant metastasis or those with a tumor measuring >2 cm accompanied by one of the following risk factors: gross ETE, age >45 years, lymph node metastasis, and distant metastasis. Otherwise, there was no significant improvement in disease-specific survival even after RAI therapy [33]. In the current study, RAI therapy was a significant risk factor of recurrence based on the multivariate analysis.

RAI therapy was conducted in patients who underwent TT and had intermediate to high risk disease according to the ATA 2015 risk stratification system for structural disease control. In other words, patients who underwent RAI therapy generally had a more advanced disease than those who did not, with larger primary tumors, more positive lymph nodes, and/or aggressive pathologic features such as vascular invasion and extrathyroidal extension. It is unlikely that RAI therapy itself increases risk for recurrence. Rather, RAI therapy may have been ineffective in reducing recurrence in patients with multifocal PTC. We surmise that this is partially due to high rate of RAI-refractory PTC. Further study should explore the efficacy of RAI therapy as adjuvant therapy for differentiated thyroid malignancies. Because RAI therapy may cause several complications such as xerostomia, sialoadenitis, and pulmonary fibrosis, cautious patient selection for RAI therapy candidates is required.

The current study had several limitations. Although the subgroup analysis aimed to eliminate the effect of surgical extent on long-term outcomes, we might have inadvertently introduced selection bias in the process of including patients with unilateral multifocal PTC who underwent TT. Large tumor size, clinical LN positivity, and other aggressive features are considered when performing TT for unilateral disease, thereby selectively including patients at a higher risk of recurrence. Further, this study was retrospective in nature and was conducted at a single center, and it included a relatively small sample size. However, all patients were treated and followed-up uniformly according to a single protocol, which is considered a strength of this research. Nevertheless, the molecular profile of the tumors, which may have important implications for clinical course and prognosis, was not investigated. Correlating the findings of our study with the standard immunohistochemical markers for aggressiveness of PTC, such as Gal-3, CK-19, and HBME-1, would provide further insight into cancer laterality as a prognostic factor for multifocal PTC.

## 5. Conclusions

Compared with bilateral PTC, unilateral multifocal PTC is significantly associated with a higher recurrence risk and poorer DFS. Our findings could affect decision making regarding surgical extent in patients with unilateral multifocal PTC. For postoperative care, patients should undergo short-term follow-up to screen for recurrence. A subsequent investigation with a larger study cohort and additional information on molecular characteristics may provide further insight about the clinical course of multifocal PTC and may help in establishing optimal treatment and follow-up strategies.

## Figures and Tables

**Figure 1 cancers-15-03596-f001:**
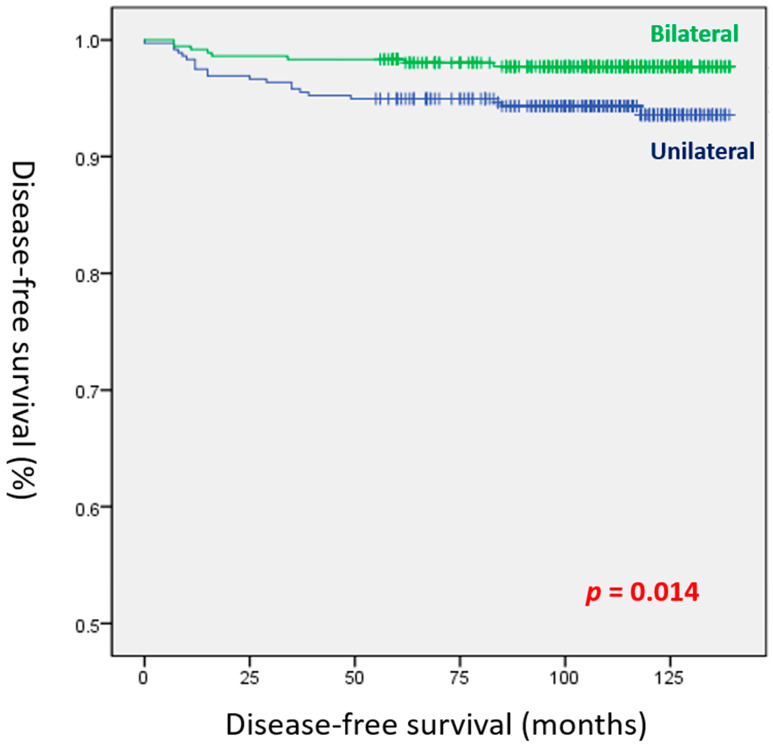
Disease-free survival curves of the bilateral and unilateral multifocal groups (log-rank test, *p* = 0.014).

**Table 1 cancers-15-03596-t001:** Comparison of baseline clinicopathological characteristics between bilateral and unilateral-multifocal PTC.

	Bilateral (*n* = 1002)	Unilateral Multifocal (*n* = 743)	*p*-Value
Age (years)	48.9 ± 12.1	47.5 ± 11.7	0.018
	(range, 15–83)	(range, 16–82)	
Male gender	199 (19.9%)	168 (22.6%)	0.172
Extent of surgery			<0.001
Less than TT	0	372 (50.1%)	
TT and/or mRND	1002 (100%)	371 (49.9%)	
Subtype of PTC			0.111
Non-aggressive	848 (84.6%)	649 (87.3%)	
Aggressive	154 (15.4%)	94 (12.7%)	
Tumor size (cm)	1.2 ± 0.9	0.9 ± 0.6	<0.001
	(range, 0.2–6.7)	(range, 0.2–6.5)	
Gross ETE	95 (9.5%)	29 (3.9%)	<0.001
Lymphatic invasion	367 (36.6%)	194 (26.1%)	<0.001
Vascular invasion	34 (3.4%)	24 (3.2%)	0.893
Perineural invasion	39 (3.9%)	10 (1.3%)	0.001
BRAFV600E positivity	712/853 (83.5%)	489/596 (82.0%)	0.479
Harvested LNs	19.1 ± 21.4	13.2 ± 16.0	<0.001
Positive LNs	3.9 ± 6.3	2.4 ± 4.5	<0.001
T stage			<0.001
T1	826 (82.4%)	689 (92.7%)	
T2	68 (6.8%)	23 (3.1%)	
T3a	13 (1.3%)	2 (0.3%)	
T3b	93 (9.3%)	27 (3.6%)	
T4a	2 (0.2%)	2 (0.3%)	
N stage			<0.001
N0	387 (38.6%)	359 (48.3%)	
N1a	455 (45.4%)	310 (41.7%)	
N1b	160 (16.0%)	74 (10.0%)	
M stage			0.402
M1	4 (0.4%)	1 (0.1%)	
TNM stage			0.002
Stage I	802 (80.0%)	643 (86.5%)	
Stage II	196 (19.6%)	100 (13.5%)	
Stage III	2 (0.2%)	0	
Stage IVb	2 (0.2%)	0	
RAI therapy	726 (72.5%)	269 (36.2%)	<0.001
Recurrence	43 (4.3%)	33 (4.4%)	0.906

Data are expressed as patient’s number (%) or mean ± standard deviation. The aggressive subtypes of PTC include tall cell, columnar, hobnail, diffuse sclerosing, and solid variant. A statistically significant difference was defined as *p* < 0.05. Abbreviation: PTC, papillary thyroid carcinoma; TT, total thyroidectomy; mRND, modified radical neck dissection; ETE, extrathyroidal extension; LN, lymph node; T, tumor; N, node; M, metastasis; RAI, radioactive iodine.

**Table 2 cancers-15-03596-t002:** Univariate and multivariate analyses of recurrence risk factors.

	Univariate		Multivariate	
	HR (95% CI)	*p*-Value	HR (95% CI)	*p*-Value
Gender				
Female	ref.			
Male	1.770 (1.091–2.870)	0.021		
Extent of surgery				
Less than TT	ref.			
TT and/or mRND	1.408 (0.760–2.610)	0.277		
Subtype of PTC				
Non-aggressive	ref.			
Aggressive	1.523 (0.866–2.681)	0.144		
Tumor size				
≤1 cm	ref.			
>1 cm	2.877 (1.821–4.547)	<0.001		
Gross ETE	3.376 (1.919–5.940)	<0.001	12.674 (1.6119–99.713)	0.016
Multifocality				
Unilateral	ref.			
Bilateral	0.953 (0.605–1.500)	0.835		
Lymphatic invasion	3.738 (2.345–5.957)	<0.001		
Vascular invasion	3.581 (1.721–7.451)	0.001		
Perineural invasion	5.011 (2.499–10.050)	<0.001	2.273 (1.005–5.139)	0.049
Harvested LNs	1.017 (1.010–1.024)	<0.001		
Positive LNs	1.078 (1.060–1.096)	<0.001	1.102 (1.053–1.154)	<0.001
T stage				
T1	ref.			
T2	2.741 (1.301–5.776)	0.008		
T3a	4.257 (1.036–17.488)	0.044		
T3b	3.635 (2.012–6.568)	<0.001		
T4b	9.142 (1.261–66.257)	0.029		
N stage				
N0	ref.		ref.	
N1a	7.535 (3.215–17.662)	<0.001	2.303 (0.912–5.819)	0.078
N1b	14.065 (5.770–34.286)	<0.001	3.565 (1.086–11.710)	0.036
M stage	10.883 (2.668–44.389)	0.001		
RAI therapy	18.685 (5.880–59.289)	<0.001	7.760 (2.275–26.470)	0.001

Data are expressed as hazard ratio (HR) and 95% confidence interval (CI). The aggressive subtypes of PTC include tall cell, columnar, hobnail, diffuse sclerosing, and solid variant. A statistically significant difference was defined as *p* < 0.05. Abbreviations: TT, total thyroidectomy; mRND, modified radical neck dissection; PTC, papillary thyroid carcinoma; ETE, extrathyroidal extension; LN, lymph node; T, tumor; N, node; M, metastasis; RAI, radioactive iodine.

**Table 3 cancers-15-03596-t003:** Comparison of baseline clinicopathological characteristics between bilateral and unilateral-multifocal PTC in patients who underwent total thyroidectomy before and after propensity score matching.

	Before Matching			After Matching		
	Bilateral (*n* = 1002)	Unilateral Multifocal (*n* = 371)	*p*-Value	Bilateral (*n* = 357)	Unilateral Multifocal (*n* = 357)	*p*-Value
Age (years)	48.9 ± 12.1 (range, 15–83)	48.1 ± 12.2 (range, 20–82)	0.305	46.5 ± 12.4 (range, 14–79)	46.2 ± 11.8 (range, 11–81)	0.793
Male gender	199 (19.9%)	89 (24.0%)	0.101	83 (23.2%)	83 (23.2%)	1.000
Subtype of PTC			0.933			0.336
Non-aggressive	848 (84.6%)	315 (84.9%)		311 (87.1%)	301 (84.3%)	
Aggressive	154 (15.4%)	56 (15.1%)		46 (12.9%)	56 (15.7%)	
Tumor size (cm)	1.2 ± 0.9 (range, 0.2–6.7)	1.0 ± 0.7 (range, 0.2–6.5)	<0.001	1.0 ± 0.6 (range, 0.2–4.5)	1.0 ± 0.7 (range, 0.2–6.5)	0.893
Gross ETE	95 (9.5%)	21 (5.7%)	0.028	22 (6.2%)	21 (5.9%)	1.000
Lymphatic invasion	367 (36.6%)	135 (36.4%)	0.950	128 (35.9%)	126 (35.3%)	0.938
Vascular invasion	34 (3.4%)	11 (3.0%)	0.865	10 (2.8%)	11 (3.1%)	1.000
Perineural invasion	39 (3.9%)	8 (2.2%)	0.134	14 (3.9%)	8 (2.2%)	0.279
BRAF^V600E^ positivity	712/853 (83.5%)	254/307 (82.7%)	0.789	255/298 (85.6%)	248/298 (83.2%)	0.498
Harvested LNs	19.1 ± 21.4	19.2 ± 20.5	0.986	19.1 ± 21.4	19.2 ± 20.5	0.463
Positive LNs	3.9 ± 6.3	3.9 ± 5.7	0.824	3.9 ± 6.3	3.9 ± 5.7	0.781
T stage			0.015			0.987
T1	826 (82.4%)	333 (89.7%)		318 (89.1%)	319 (89.4%)	
T2	68 (6.8%)	16 (4.3%)		16 (4.5%)	16 (4.5%)	
T3a	13 (1.3%)	1 (0.3%)		1 (0.3%)	1 (0.3%)	
T3b	93 (9.3%)	20 (5.4%)		20 (5.6%)	20 (5.6%)	
T4a	2 (0.2%)	1 (0.3%)		2 (0.6%)	1 (0.3%)	
N stage			0.189			0.842
N0	387 (38.6%)	131 (35.3%)		128 (35.9%)	129 (36.1%)	
N1a	455 (45.4%)	166 (44.8%)		157 (44.0%)	162 (45.4%)	
N1b	160 (16.0%)	74 (19.9%)		72 (20.2%)	66 (18.5%)	
M stage			0.579			N/A
M1	4 (0.4%)	0		0	0	
TNM stage			0.391			0.303
Stage I	802 (80.0%)	309 (83.3%)		288 (80.7%)	296 (82.9%)	
Stage II	196 (19.6%)	62 (16.7%)		67 (18.8%)	61 (17.1%)	
Stage III	2 (0.2%)	0		2 (0.6%)	0	
Stage IVb	2 (0.2%)	0		0	0	
RAI therapy	726 (72.5%)	251 (67.7%)	0.093	246 (68.9%)	242 (67.8%)	0.809
Recurrence	43 (4.3%)	21 (5.7%)	0.313	8 (2.2%)	21 (5.9%)	0.021

Data are expressed as patient’s number (%) or mean ± standard deviation. The aggressive subtypes of PTC include tall cell, columnar, hobnail, diffuse sclerosing, and solid variant. A statistically significant difference was defined as *p* < 0.05. The *p*-value for M stage is unavailable since none of the patients had M1 disease after matching. Abbreviation: PTC, papillary thyroid carcinoma; ETE, extrathyroidal extension; LN, lymph node; T, tumor; N, node; M, metastasis; RAI, radioactive iodine.

**Table 4 cancers-15-03596-t004:** Univariate and multivariate analyses of recurrence risk factors after propensity score matching.

	Univariate		Multivariate	
	HR (95% CI)	*p*-Value	HR (95% CI)	*p*-Value
Tumor size				
≤1 cm	ref.			
>1 cm	2.102 (1.014–4.358)	0.046		
gross ETE	3.458 (1.318–9.070)	0.012		
Multifocality				
Bilateral	ref.		ref.	
Unilateral	2.660 (1.178–6.004)	0.019	2.664 (1.180–6.017)	0.018
Lymphatic invasion	3.586 (1.667–7.712)	0.001		
Vascular invasion	5.768 (2.006–16.582)	0.001	3.839 (1.331–11.073)	0.013
Perineural invasion	3.882 (1.175–12.833)	0.026		
Positive LNs	1.057 (1.023–1.093)	0.001		
T stage				
T1	ref.			
T3b	3.041 (1.047–8.830)	0.041		
T4b	12.909 (1.727–96.474)	0.013		
N stage				
N0	ref.			
N1a	4.437 (1.293–15.228)	0.018		
N1b	6.678 (1.836–24.288)	0.004		
RAI therapy	13.334 (1.814–98.005)	0.011	12.124 (1.640–89.630)	0.014

Data are expressed as hazard ratio (HR) and 95% confidence interval (CI); a statistically significant difference was defined as *p* < 0.05. Abbreviations: ETE, extrathyroidal extension; LN, lymph node; T, tumor; N, node; RAI, radioactive iodine.

## Data Availability

The data supporting the findings of this study are available upon request from the corresponding author and are not publicly available due to privacy or ethical restrictions.

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
