# Peer review of "Clinical Implication of Bilateral and Unilateral Multifocality in Papillary Thyroid Carcinoma: A Propensity Score-Matched Study"

_cancers, 2023, doi:10.3390/cancers15143596_

Round 1

Reviewer 1 Report

This is a thorough study comparing the outcome and clinicopathological characteristics of unilateral and bilateral papillary thyroid carcinomas. The study is well designed and the conclusions seem sound. However, I have some serious concerns especially in the Results section.

Minor concerns:

1. The authors should explain in more details how PSM is conducted

2. BRAF genotypisation is not described

3. It should be explained how RAI can pose a risk factor for recurrence

4. Can the authors correlate their findings with standard immunohistochemical markers of aggressiveness of PTC (Gal-3, CK-19, HBME-1)

5. Lines 196-206 should be moved to the Introduction section

6. Lines 52 and 95 are incomplete sentences

7. What is modified radical neck resection?

Major concerns:

1. In Table 1, the ranges for age and tumor size are unclear. Ranges for age are shown in two rows (15-83, an 16-82). For tumor size, it is the same situation (0.2-6.7 and 0.2-6.5)

2. The p-value for age in Table 1 is misplaced

3. Why are there no data on age and tumor size for Unilateral-multifocal tumors

4. In Table 2, there are 4 patients staged TNM III and TNM IVb. Why are these patients not listed in Table1?

5. The total number of cases is not uniform across tables. For example, the sum of cases with Lymphatic invasion, Vascular invasion and Perineural invasion is 440 in bilateral PTC, not 1002. There are multiple such mistakes. Please revise the tables.

Moderate changes required

Author Response

Response to Reviewer 1 Comments

Minor concerns:

Point 1: The authors should explain in more details how PSM is conducted

Response 1: To reduce the impact of selection bias and potential confounding factors, propensity score matching was performed using seven clinicopathological and biochemical characteristics: sex, tumor size, gross extrathyroidal extension, perineural invasion, T and N stages, and RAI therapy.

Point 2: BRAF genotypisation is not described

Response 2: Testing for BRAF V600E mutation was carried out via real-time PCR of tumor specimen acquired by either fine-needle biopsy or surgical resection, as well as BRAF V600E-specific staining of tumors. A positive result was considered positive for BRAF mutation. We had not included the methods for BRAF genotypisation in our original manuscript since we did not consider it as the main focus of our study. We have now added BRAF genotypisation in the methods section. We thank you for your detailed review of our study.

Point 3: It should be explained how RAI can pose a risk factor for recurrence

Response 3: In our original manuscript, we had not explained clearly how we interpreted the results on RAI therapy as a risk factor for recurrence. We believe it is unlikely that RAI therapy itself increases risk for recurrence; rather, we question its effect as adjuvant therapy to reduce recurrence. In our study, patients who underwent postoperative RAI therapy had intermediate to high risk disease according to the ATA 2015 risk stratification system for structural disease recurrence. By definition, those patients were at a higher risk of recurrence than patients who had low risk disease and did not undergo RAI therapy. Therefore, we surmised that RAI therapy failed to show protective effect on recurrence, partially owing to high rate of RAI-refractory PTC. Further study should explore the efficacy of RAI therapy as adjuvant therapy for differentiated thyroid malignancies. We have elaborated on our interpretation of the results in the manuscript. We thank you for your comment that improved our study.

Point 4: Can the authors correlate their findings with standard immunohistochemical markers of aggressiveness of PTC (Gal-3, CK-19, HBME-1)

Response 4: Unfortunately, not all the included patients were tested for the abovementioned immunohistochemical markers, so we are unable to make such correlations at this time. We believe that correlating our findings with IHC markers for aggressiveness would have strengthened our study as you have suggested. We will include it as a limitation of our study. We appreciate your insightful comment.

Point 5: Lines 196-206 should be moved to the Introduction section

Response 5: The second paragraph (lines 196-206) is essentially the background information of this study, which should be placed in the Introduction, as you have pointed out. Since similar contents are already included in the Introduction section, we simply shortened the paragraph to remind the essence of this study before we move on to explain the findings of our study. We thank you for improving our work.

Point 6: Lines 52 and 95 are incomplete sentences

Response 6: The beginning of each sentence was omitted by accident. Corrections were made for both sentences. We thank you for pointing out what we missed.

Point 7: What is modified radical neck resection?

Response 7: Modified radical neck dissection (mRND) is defined as removal of cervical lymph nodes in level I ~ V as in radical neck dissection, but preserving at least one of the ipsilateral non-lymphatic structures such as sternocleidomastoid muscle, spinal accessory nerve, and internal jugular vein. MRND was performed in patients who were preoperatively diagnosed with metastatic lymph node(s) in the lateral neck via fine needle aspiration biopsy. We thank you for improving the details of our study.

Major concerns:

Point 1: In Table 1, the ranges for age and tumor size are unclear. Ranges for age are shown in two rows (15-83, an 16-82). For tumor size, it is the same situation (0.2-6.7 and 0.2-6.5)

Response 1: The values and ranges for age and tumor size were misplaced on the left column altogether, under the “bilateral” column. We have made the appropriate corrections to Table 1. We thank you for pointing out what we missed.

Point 2: The p-value for age in Table 1 is misplaced

Response 2: We have made the correction as you pointed out. We thank you for pointing out what we missed.

Point 3: Why are there no data on age and tumor size for Unilateral-multifocal tumors

Response 3: The data on age and tumor size for unilateral-multifocal tumors were shifted to the left column, under the bilateral data, as mentioned in the response for major point 1. We have made the appropriate changes to Table 1. We thank you for pointing out what we missed.

Point 4: In Table 2, there are 4 patients staged TNM III and TNM IVb. Why are these patients not listed in Table1?

Response 4: The data was omitted from the table by accident. We have added the data for stages III, IVb in Table 1. We thank you for your detailed review of our work.

Point 5: The total number of cases is not uniform across tables. For example, the sum of cases with Lymphatic invasion, Vascular invasion and Perineural invasion is 440 in bilateral PTC, not 1002. There are multiple such mistakes. Please revise the tables.

Response 5: The cases for lymphatic invasion, vascular invasion and perineural invasion do not sum up to 1002 for the bilateral group, because not all patients had such features. Instead, the percentage of patients with each pathologic feature is noted in the table. Also, one patient may be included multiple times, if he or she has lymphatic/vascular/perineural invasions concurrently. Additionally, not all patients had molecular testing for BRAFV600E. Therefore, the number of patients who underwent molecular testing was divided by the number of patients who tested positive for BRAFV600E mutation to yield an appropriate percentage. Otherwise, the number of patients under “extent of surgery,” “subtype of PTC,” “T/N stages,” “TNM stage” should all sum up to 1002 for the bilateral group and 743 for the unilateral-multifocal group. We appreciate your detailed review of our study.

Reviewer 2 Report

I analyzed the manuscript entitled “Clinical Implication of Bilateral and Unilateral Multifocality in Papillary Thyroid Carcinoma: A Propensity Score-Matched Study”, who aims to evaluate the multifocal bilateral and unilateral PTCs by comparing the clinicopathological characteristics and long term oncological characteristics in a propensity score-matched study. 

Although it is of major importance for surgeons and oncologists, this issues was not approached by many studies before. Previous studies compared risk of recurrence and disease-free survival between multifocal vs. unifocal PTCs, but only few compared multifocal bilateral vs. multifocal unilateral. These aspects, together with the propensity score-matched fashion in designing the study group bring originality and strength to the present study. 

However, there are some minor concerns regarding the methodology of the research, as follows: 
- The authors should explain why the T1 patients were analyzed in a single group and not separately, T1a and T1b subgroups.
- In Table 3, data regarding Recurrences (last line) “After matching”, should be checked with the data in the main text (line 164).
- It would be of interest to compare the tumors with subcapsular or intraparenchymal location, as this might have consequences on the infiltrative behavior and recurrences. 
- The authors should mention the references used for the TNM classification.  The conclusions are pertinent, sustained by the study’s results and according to the aim of the research.   Should these concerns be properly addressed, I would recommend this manuscript for publication.

Author Response

Response to Reviewer 2 Comments

Point 1: The authors should explain why the T1 patients were analyzed in a single group and not separately, T1a and T1b subgroups.

Response 1: Since the 7th edition of the AJCC pTNM staging system, T1 category was subdivided into T1a(≤10mm) and T1b(11-20mm), which was preserved in the 8th edition. However, the prognostic value of this subdivision remains controversial [1,2]. Moreover, a study by Tam et al. found that total tumor diameter (TTD) of multifocal disease, calculated as the sum of the maximal diameter of each lesion, affects prognosis. They found that TTD >10mm in multifocal PTC yielded a similar risk of aggressive behavior as T1b unifocal disease [3]. In the current study, only T3b and T4b stages were significant factors for recurrence on univariate analysis after PSM. Therefore, we decided to analyze the T1 group as a single group instead of subdividing it into two groups. We appreciate your insightful comment.

Point 2: In Table 3, data regarding Recurrences (last line) “After matching”, should be checked with the data in the main text (line 164).

Response 2: We have revised the text with the correct values shown in Table 3: “After PSM, the unilateral multifocal PTC group had a higher recurrence rate than the bilateral PTC group (2.2% vs. 5.9%, p = 0.021).” We thank you for pointing out what we missed.

Point 3: It would be of interest to compare the tumors with subcapsular or intraparenchymal location, as this might have consequences on the infiltrative behavior and recurrences.

Response 3: We agree that comparison of tumor behavior and recurrence rate according to the tumor location would provide valuable insight into understanding multifocal PTC further. We are unable to identify the location of primary tumors in this study since the patient identification data have undergone anonymization. However, we would like to explore the clinical implication of the primary tumor location in our future studies. We thank you for your thoughtful contribution.

Point 4: The authors should mention the references used for the TNM classification. 

Response 4: We have applied 8th Ed. AJCC staging system for TNM classification. We have added the reference as you have recommended. We appreciate your detailed review of our study.

Reference

  1. Wang, L.Y.; Nixon, I.J.; Palmer, F.L.; Thomas, D.; Tuttle, R.M.; Shaha, A.R.; Patel, J.P.; Ganly, I. Comparable outcomes for patients with pT1a and pT1b differentiated thyroid cancer: Is there a need for change in the AJCC classification system? Surgery 2014, 156, 1484-1489.
  2. Chereau, N.; Tresallet, C.; Noullet, S.; Godiris-Petit, G.; Tissier, F.; Leenhardt, L.; Menegaux, F. Does the T1 subdivision correlate with the risk of recurrence of papillary thyroid cancer? Langenbecks Arch Surg 2016, 401, 223-230.
  3. Tam, A.A.; Ozdemir, D.; Cuhaci, N.; Baser, H.; Aydm, C.; Yazgan, A.K.; Ersoy, R.; Cakir, B. Association of multifocality, tumor number, and total tumor diameter with clinicopathological features in papillary thyroid cancer. Endocrine 2016, 53, 774-783.
  4. Tuttle, M.; Morris, L.F.; Haugen, B.; etc. 2017 Thyroid differentiated and anaplastic carcinoma. In AJCC Cancer Staging Manual, 8th Ed; Amin, M.B., Edge, S.B., Eds.; Springer International Publishing: New York, USA, 2017, ISBN978-3319406176.

Reviewer 3 Report

There are no significant remarks to the manuscript, except for the Discussion, which will be written below.

A few minor notes:

Lines 38-39: “The incidence of multifocal PTC ranges from 18% to 87% [4,5,6]. PTC presents as a multifocal disease at diagnosis.”

There is an impression that something is missing here, because the sentences do not agree well with each other.

Line 52: , some studies…”

Unexpectedly, the sentence starts with a comma.

Line 95: “were analyzed with the Student’s…”

Apparently, something is missing at the beginning of the sentence.

Line 103: “…PTV were matched to those with bilateral PTC…”

“PTV” is, as I understand it, also PTC.

Line 113: “The two groups did not significantly differ in terms of age and sex.”

If I understood Table 1 correctly, then for Age p=0.018 < 0.05, which means a significant difference.

Line 118: “…p < 0.001…”

Here and below, the actual value of p should be given, as this may be of interest to readers.

Table 1.

The Age (years) and Tumor size (cm) items look wrong in the table, because all the values are collected in the Bilateral row.

Discussion

Lines 194-227: The first 4 paragraphs are essentially a second introduction, not a discussion, since the results obtained in this study appear only in the 5th paragraph. Thus, the first four paragraphs should be moved to the Introduction, or simply shortened a lot to remind the essence of the matter without going into details.

Further, in this work, results were obtained that partially contradict what was previously considered, and as I understand it, this is due to the use of propensity score matching (PSM). For this reason, the discussion should pay more attention to the PSM and the possible errors that can occur if this correction is ignored.

Finally, a discussion should be added on how the result can be explained - that unilateral multifocal PTC, compared with bilateral PTC, is significantly associated with a higher recurrence risk and poorer DFS. For example, because bilateral multifocality is more likely to be associated with the independent development of tumors, while unilateral multifocality is more likely to be associated with the active spread of one tumor within the lobe, which may indicate its greater aggressiveness. Or any other ideas that the authors have on this issue.

Author Response

Response to Reviewer 3 Comments

There are no significant remarks to the manuscript, except for the Discussion, which will be written below.

A few minor notes:

Point 1: Lines 38-39: “The incidence of multifocal PTC ranges from 18% to 87% [4,5,6]. PTC presents as a multifocal disease at diagnosis.”

There is an impression that something is missing here, because the sentences do not agree well with each other.

Response 1: The sentence has been revised for better flow of the text: “The incidence of multifocal PTC ranges from 18% to 87% [4,5,6]. It is not uncommon for PTC to present as a multifocal disease at diagnosis.” We appreciate your insightful comment.

Point 2: Line 52: “, some studies…”

Unexpectedly, the sentence starts with a comma.

Response 2: A conjunctive adverb was omitted by accident, which we have added: “On the contrary, some studies…” We appreciate your comment for correcting what we missed.                                                                                                                                         

Point 3: Line 95: “were analyzed with the Student’s…”

Apparently, something is missing at the beginning of the sentence.

Response 3: The beginning of the sentence had been omitted. We have corrected the sentence: “Continuous variables were analyzed with the Student’s t-test…” We thank you for pointing out what we missed.

Point 4: Line 103: “…PTV were matched to those with bilateral PTC…”

“PTV” is, as I understand it, also PTC.

Response 4: Correction was made accordingly: “PTC.” We appreciate your detailed review of our study.

Point 5: Line 113: “The two groups did not significantly differ in terms of age and sex.”

If I understood Table 1 correctly, then for Age p=0.018 < 0.05, which means a significant difference.

Response 5: We agree that there is a significant difference of age between the two groups. The text has been revised accordingly: “The mean age of the bilateral PTC group was significantly higher than that of the unilateral-multifocal PTC group, while the two groups did not show difference in sex.” We thank you for pointing out an important correction to be made. Our work has been improved as the result.

Point 6: Line 118: “…p < 0.001…”

Here and below, the actual value of p should be given, as this may be of interest to readers.

Response 6: The p-values have been added to the manuscript according to your recommendation. We thank you for your insightful comment.

Point 7: Table 1: The Age (years) and Tumor size (cm) items look wrong in the table, because all the values are collected in the Bilateral row.

Response 7: The table has been revised with values in the correct row. We thank you for pointing out what we missed.

Discussion

Point 8: Lines 194-227: The first 4 paragraphs are essentially a second introduction, not a discussion, since the results obtained in this study appear only in the 5th paragraph. Thus, the first four paragraphs should be moved to the Introduction, or simply shortened a lot to remind the essence of the matter without going into details.

Response 8: In the first paragraph, we aimed to emphasize how the findings of our study differs from those of previous reports. The following 2nd ~ 4th paragraphs were shortened into a single paragraph. We appreciate your comment that helped us improve our work.

Point 9: Further, in this work, results were obtained that partially contradict what was previously considered, and as I understand it, this is due to the use of propensity score matching (PSM). For this reason, the discussion should pay more attention to the PSM and the possible errors that can occur if this correction is ignored.

Response 9: We have elaborated on the effect of PSM on our results: “PSM yielded rather an unexpected outcome. Before matchcing, the bilateral and the unilateral multifocal PTC groups did not show a significant difference in recurrence, which became significant once PSM was conducted. It suggests that the clinicopathologic characteristics, such as tumor size and gross extrathyroidal extension, play an important role in determining the oncological outcome. Therefore, it is important to correct for the possibly confounding factors in order to make an accurate comparison of cancer laterality regarding prognosis.” We thank you for your insightful comment that helped us strengthen our manuscript.

Point 10: Finally, a discussion should be added on how the result can be explained – that unilateral multifocal PTC, compared with bilateral PTC, is significantly associated with a higher recurrence risk and poorer DFS. For example, because bilateral multifocality is more likely to be associated with the independent development of tumors, while unilateral multifocality is more likely to be associated with the active spread of one tumor within the lobe, which may indicate its greater aggressiveness. Or any other ideas that the authors have on this issue.

Response 10: We agree that unilateral-multifocality may be more aggressive in nature since it possibly arises from intraglandular tumor dissemination, compared to bilaterality, which is more often associated with multiple synchronous primary tumors (MSPT). We had previously cited a work by Cai et al, which suggests unilateral multifocality and bilateral multifocality as molecularly independent entities. We have further elaborated on the subject with an additional reference. We sincerely appreciate your review for further improving our work.

Round 2

Reviewer 1 Report

The manuscript has now been sufficiently improved to be suitable for publication